# Improved Accuracy and Sensitivity in Diagnosis and Staging of Lung Cancer with Systematic and Combined Endobronchial and Endoscopic Ultrasound (EBUS-EUS): Experience from a Tertiary Center

**DOI:** 10.3390/cancers16040728

**Published:** 2024-02-09

**Authors:** Abdenor Badaoui, Marion De Wergifosse, Benoit Rondelet, Pierre H. Deprez, Claudia Stanciu-Pop, Laurent Bairy, Philippe Eucher, Monique Delos, Sebahat Ocak, Cédric Gillain, Fabrice Duplaquet, Lionel Pirard

**Affiliations:** 1Department of Gastroenterology and Hepatology, CHU UCL Namur, Université catholique de Louvain, 5530 Yvoir, Belgium; abdenor.badaoui@chuuclnamur.uclouvain.be (A.B.); cedric.gillain@chuuclnamur.uclouvain.be (C.G.); 2Department of Pneumology, CHU UCL Namur, Université catholique de Louvain, 5530 Yvoir, Belgium; marion.dewergifosse@student.uclouvain.be (M.D.W.); sebahat.ocak@chuuclnamur.uclouvain.be (S.O.); fabrice.duplaquet@chuuclnamur.uclouvain.be (F.D.); 3Department of Vascular and Thoracic Surgery, CHU UCL Namur, Université catholique de Louvain, 5530 Yvoir, Belgium; benoit.rondelet@chuuclnamur.uclouvain.be (B.R.); philippe.eucher@chuuclnamur.uclouvain.be (P.E.); 4Department of Gastroenterology and Hepatology, Cliniques Universitaires Saint-Luc, Université catholique de Louvain, 1200 Bruxelles, Belgium; pdeprez@saintluc.uclouvain.be; 5Department of Pathological Anatomy, CHU UCL Namur, Université catholique de Louvain, 5530 Yvoir, Belgium; cpop@chuliege.be (C.S.-P.); monique.delos@chuuclnamur.uclouvain.be (M.D.); 6Department of Anesthesiology, CHU UCL Namur, Université catholique de Louvain, 5530 Yvoir, Belgium; laurent.bairy@chuuclnamur.uclouvain.be; 7Pole of Pneumology, Institut de Recherche Expérimentale et Clinique, Université catholique de Louvain, 1200 Bruxelles, Belgium

**Keywords:** mediastinal staging, lung cancer, endoscopic ultrasound, endobronchial ultrasound, mediastinoscopy

## Abstract

**Simple Summary:**

Lung cancer represents the most common form of cancer worldwide and the most frequent cause of cancer-related death in men and women combined. Lung cancer staging is very important, especially in patients who could benefit from surgery. Endobronchial ultrasound (EBUS) and endoscopic ultrasound (EUS) are complementary techniques to explore and acquire tissue from mediastinal lymph nodes by trans-tracheal/bronchial and trans-esophageal approaches, respectively. The respective contribution of separate and combined procedures in the diagnosis and staging of lung cancer has not been fully studied. In our study, a total of 141 patients underwent both procedures, and the combined EBUS-EUS approach in lung cancer patients showed better accuracy and sensitivity in the diagnosis and staging of lung cancer when compared with EBUS and EUS alone. It demonstrated the unmissable aspect of the systematic combination of these endosonographic techniques for an optimal mediastinal diagnosis and staging in lung cancer for patients’ survival.

**Abstract:**

Background: Combined endobronchial ultrasound-guided transbronchial needle aspiration (EBUS-TBNA) and endoscopic ultrasound-guided tissue acquisition (EUS-TA) are accurate procedures for the diagnosis and staging of mediastinal lymph nodes (MLNs) in lung cancer. However, the respective contribution of separate and combined procedures in diagnosis and staging has not been fully studied. The aim of this study was to assess their respective performances. Methods: Patients with suspected malignant MLNs in lung cancer or recurrence identified by PET-CT who underwent combined EBUS-TBNA and EUS-TA were retrospectively reviewed. Results: A total of 141 patients underwent both procedures. Correct diagnosis was obtained in 82% with EBUS-TBNA, 91% with EUS-TA, and 94% with the combined procedure. The overall sensitivity, specificity, and positive and negative predictive values (PPV and NPV) of EBUS-TBNA, EUS-TA, and the combined procedure for diagnosing malignancy were [75%, 100%, 100%, 58%], [87%, 100%, 100%, 75%], and [93%, 100%, 100%, 80%], respectively, with a significantly better sensitivity of the combined procedure (*p* < 0.0001). Staging (82/141 patients) was correctly assessed in 74% with EBUS-TBNA, 68% with EUS-TA, and 85% with the combined procedure. The overall sensitivity, specificity, PPV, and NPV of EBUS-TBNA, EUS-TA, and the combined procedure for lung cancer staging were [62%, 100%, 100%, 55%], [54%, 100%, 100%, 50%], and [79%, 100%, 100%, 68%], respectively, significantly better in terms of sensitivity for the combined procedure (*p* < 0.001). Conclusion: The combined EBUS-EUS approach in lung cancer patients showed better accuracy and sensitivity in diagnosis and staging when compared with EBUS-TBNA and EUS-TA alone.

## 1. Introduction

Lung cancer represents the most common form of cancer worldwide and the most frequent cause of cancer-related death in men and women combined, causing an estimated 1.8 million deaths in 2018. In Europe, it is also the leading cause of cancer-related deaths, with approximately 388,000 deaths in 2018, representing 1 in 5 of all cancer deaths. It is also the second most common cancer [1].

In lung cancer, long-term survival of patients with cN2 disease who could benefit from surgery, possibly after neoadjuvant therapy, varies markedly according to their preoperative characteristics and the extent of preoperative staging investigations. Although mediastinoscopy is the gold standard in lung cancer staging, its indications have considerably decreased in the last decade with the performance of endosonography in MLN analysis and staging. European guidelines recommend endosonography over surgical staging as the first step for mediastinal staging in patients with suspected or proven non-small-cell lung cancer (NSCLC) with abnormal mediastinal and/or hilar lymph nodes on computed tomography (CT) and/or positron emission tomography (PET) [2,3].

EBUS and EUS are complementary techniques to explore and acquire tissue from MLNs using trans-tracheal/bronchial (EBUS-TBNA) and trans-esophageal (EUS-TA) approaches, respectively. They reach common and different MLN stations [2,3,4,5,6,7,8]. The European guidelines published in 2015 recommend combining EBUS and EUS for mediastinal staging of lung cancer, and, if the combination is not possible, to perform EBUS preferably (Grade C) [2,3]. Combined EBUS-EUS is indeed less invasive than mediastinoscopy and avoids unnecessary thoracotomies [4,5,9,10]. However, the respective contribution of separate and combined procedures in the diagnosis and staging of MLNs in lung cancer patients has not been fully studied. The aim of this study was to assess their respective performances.

## 2. Materials and Methods

### 2.1. Study Design and Subjects

Data were collected from a prospective registry of patients established between December 2009 and October 2020, then analyzed retrospectively. The 141 patients who were included had MLNs within a framework of suspected or confirmed lung cancer and underwent combined EBUS-TBNA and EUS-TA (Figure 1). A total of 108 were outpatients and the remaining patients were hospitalized. The study was approved by the local ethical committee and was performed in accordance with the Declaration of Helsinki. Informed consent from individual patients was not required by the ethical committee given the retrospective nature of the study.

### 2.2. Procedures

Previous diagnostic flexible bronchoscopy was performed in all patients. The combined procedure was performed under general anesthesia. EBUS and EUS were performed in a single-step procedure or separately on different days using linear-array echoendoscopes (Olympus UC-180F for EBUS (Tokyo, Japan) and Pentax EG 3830UT (Tokyo, Japan) or Olympus GF-UCT 180 for EUS (Tokyo, Japan)).

First, a laryngeal mask was placed on the patient for the EBUS procedure, which was performed by a pulmonologist. A 21/22-g needle (Olympus) was used for the puncture of targeted mediastinal stations with 3 to 4 passages after a complete examination of the anterior and lateral mediastinum.

Secondly, the patient was intubated and the EUS procedure was performed by a gastroenterologist. After complete examination of the posterior mediastinum, puncture of the targeted MLN was carried out with fine needle aspiration (FNA) 22-g (Boston Scientific or Cook Medical) or fine needle biopsy (FNB) 22-g or 20-g (Boston Scientific or Cook Medical), with 2 to 3 passages. The samples were collected and placed in CytoLyt solution (Thinprep^®^ Cytolyt Solution-Hologic).

After a minimum of two hours of fixation in Cytolyt, the cytology was centrifugated and the supernatant was discarded. Two droplets of material were placed in a vial containing PreservCyt solution (Hologic, Marlborough, MA, USA) to be processed in a Thinprep 2000 processor (Hologic). One Papanicolaou-stained monolayer was available. The rest of the material was embedded in Agar and paraffin-embedded. One 4 µm thick hematoxylin and eosin (H&E)-stained slide from the cell block was also available for the diagnosis. The cell block was used for the immunohistochemistry and also for the molecular biology.

Mediastinitis, pulmonary infection, severe hemorrhage, and esophageal perforation were considered as serious complications.

In lung cancer staging, patients with benign MLNs diagnosed after combined EBUS-EUS underwent mediastinoscopy and/or surgery with lymphadenectomy (SLA) (SLA was performed in some cases without mediastinoscopy when this was not feasible for technical and anatomical reasons). In rare situations, follow-up was indicated according to the patient’s history and condition. Video-assisted mediastinoscopy (VAM), which improves the visualization of lymph nodes (LNs) [6], was mostly performed rather than classical mediastinoscopy, and similarly, SLA was most frequently performed with video-assisted thoracic surgery (VATS) rather than with thoracotomy.

### 2.3. Statistical Analysis

The accuracy, sensitivity, specificity, PPV, and NPV for MLN diagnosis of malignancy with or without staging of lung cancer with EBUS-TBNA, EUS-TA, and the combined procedure were calculated with a 95% confidence interval (CI), and then comparisons were performed using McNemar’s test. A *p* value of less than 0.05 was considered as significantly different.

From a sample size of 141 patients, by simulation and considering a power of 90% and paired data, we noted a significant difference in sensitivity of 7% between both EBUS-TBNA and EUS-TA, which were carried out separately, and the combined procedure, which is considered as relevant.

Considering the 82 patients who had endosonographic staging and the same statistical hypotheses, we noted a significant difference of 12% in sensitivity between both EBUS-TBNA and EUS-TA, when performed separately, and the combined procedure, which is considered as relevant.

The software program R 4.0.1 (R Foundation for Statistical Computing, Vienna, Austria) with the epiR package was used.

The gold standard indicated a positive result for the malignancy results obtained with the combined procedure, or malignant MLNs found after combined EBUS-TBNA-EUS-TA and mediastinoscopy (conventional mediastinoscopy or VAM), or with SLA when the combined procedure showed N0/N1 stage with PET-CT revealing suspected MLNs if potential surgery was considered (SLA), or patients with N2 stage with combined procedure who would benefit from SLA or need a confirmation by mediastinoscopy before chemoradiotherapy. Negative results for malignancy after combined procedure had to be confirmed by mediastinoscopy, SLA or follow-up (Figure 1).

After mediastinal staging with EBUS-TBNA and EUS-TA, statuses N1, N2, and N3 were considered as true positives when confirmed via final staging, while N0 status was considered as a true negative. When N0, N1, and N2 stages obtained using the combined procedure were upstaged in the final results, they were considered as false negatives.

## 3. Results

A total of 141 patients with suspected malignant MLNs with known or suspected lung cancer, as identified by PET-CT, underwent systematic combined EBUS-TBNA and EUS-TA—as a single procedure in 129 patients and in two procedures in 12 patients (mean period between procedures 14 days, range 4–26). The EBUS and EUS procedures related to these 12 cases were separately performed at the beginning of the study due to organizational considerations, but always in a systematic combined stetting, regardless of negative or positive lymph nodes for malignancy being diagnosed by one or the other procedure. Of these 141 patients, 120 had MLNs with lung lesions suspected to be cancer and 21 had a history of lung cancer with MLNs with suspected recurrence. Among the 141 patients, 59 patients underwent a systematic combined procedure for diagnosis and 82 patients underwent a systematic combined procedure for both diagnosis and staging. EUS-FNA was performed in the first 50 patients and EUS-FNB was carried out in the remaining patients. The mean duration of the procedure was 50 min (range: 37–71). One serious complication related to the EUS procedure was observed: pleural empyema in a patient who underwent a puncture of a lung lesion with EUS-FNB and which was managed by CT-scan drainage and antibiotics, with a favorable course.

Table 1 summarizes the pathological results. Most patients (85) were diagnosed with non-small-cell lung cancer (NSCLC). The most frequent MLNs reached were in the right lower paratracheal (4R) and subcarinal (7) stations with EBUS-TBNA and in the left lower paratracheal (4L) and subcarinal (7) stations with EUS-TA. Lung masses were punctured in four and six patients who received EBUS-TBNA and EUS-TA, respectively, in addition to puncture of MLNs.

Moreover, EUS-TA enabled puncturing of the left adrenal gland in 21 patients and a left hepatic lesion in 1 patient (Table 2).

The sensitivities for malignancy diagnosis of EBUS-TBNA, EUS-TA, and combined EBUS-TBNA/EUS-TA were 75% [95% CI: 66–83%], 87% [95% CI: 79–93], and 93% [95% CI: 86–97], respectively, with a significantly higher sensitivity achieved by the combined procedure compared with EBUS-TBNA (*p* < 0.0001), but not by comparing EUS-TA and the combined approach (*p* = 0.13) (Table 3).

Similarly, staging was significantly improved by the combined approach in terms of sensitivity, accuracy, and NPV. There was a statistically significant difference in sensitivity (12%) between EBUS-TBNA and the combined approach (*p* = 0.0077), and between EUS-TA and the combined approach (*p* = 0.00051) (Table 4).

Among 37 patients with benign lesions (MLN), 28 were true negatives (confirmation by mediastinoscopy (2), mediastinoscopy + SLA (11), SLA (6), or follow-up (9)). In nine patients, the results of the combined approach was considered as false negatives, with the status being upstaged by mediastinoscopy (2), mediastinoscopy + SLA (4) and SLA (2), or follow-up (1)**.** Eleven patients with positive lesions for malignancy (MLNs, for most of them, and lung lesions), which were essentially N2 and N0 after the combined procedure, underwent mediastinoscopy in order to consider SLA, their staging results being confirmed for most of them (10/11 patients) (Figure 1).

The combined procedure upstaged five patients compared to the PET-CT data, from N0 to N2, N1 to N2, N1 to N3, and N2 to N3 disease in one, one, one, and two patients, respectively. The contribution of EUS-TA identified the correct N stage in three patients from PET-CT data (shift of N-stage from N1 to N2 and N2 to N3 stages in one and two patients, respectively). From EBUS-TBNA data, EUS-TA enabled a shift in the N-stage from N2 to N3 disease in two patients, and, conversely, EBUS-TBNA enabled a shift from N0 to N2 and N1 to N3 disease in two patients from the EUS-TA data. Finally, both EBUS-TBNA and EUS-TA identified the correct N stage in the same patient from PET-CT data (N1 to N2 disease) (Figure 2).

In nine patients, EUS-TA enabled, in an unusual way, the measuring of a puncture of a suspected LN of at least 20 mm in the right and anterior lower paratracheal station 4R. In seven patients, EUS-TA was performed for diagnosis and staging, and determined a diagnosis in six patients. SLA provided the diagnosis and staging in one patient in whom the puncture of LN in the 4R station was a false negative not only in EUS-TA, but also with EBUS-TBNA. Finally, the diagnosis was obtained with EUS-TA in the two remaining patients who presented with metastatic disease (Table 2).

In one patient with a right pulmonary squamous cell carcinoma (SCC), EUS-TA and mediastinoscopy staged the patient as N0, but the patient was staged as N1 by EBUS-TBNA with a positive LN in the 11R station (inaccessible for EUS).

Three patients were identified as positive for malignancy and staged N2 with the combined procedure, but presented supraclavicular LN punctured under ultrasound guidance, which proved malignancy. Therefore, the N status was upstaged to N3 disease (considered as a false negative).

## 4. Discussion

In the last decade, diagnosis and staging for lung cancer has evolved from mediastinoscopy and surgical staging towards EBUS-TBNA, EUS-TA, or, ideally, a combination of these procedures [11,12]. Our series confirms the better staging accuracy of the combined approach compared with each procedure alone, with a statistically significant difference in terms of sensitivity (12%) between EBUS-TBNA and the combined approach (*p* < 0.01), and between EUS-TA and the combined approach (*p* < 0.001).

These results underline the importance of each technique, including EUS-TA, to the staging accuracy. The literature results are unclear about this. Oki et al. showed, in a prospective study including 150 patients with staging of potentially resectable known or suspected lung cancer, no statistical difference in terms of sensitivity between EBUS-TBNA and the combined procedure, but a significantly higher sensitivity of the combined procedure compared with EUS-FNA [13].

Lee et al. retrospectively showed, in 37 patients, a higher sensitivity of mediastinal staging with the combined procedure, with a significant difference between EBUS-TBNA and the combined procedure, but the difference between EUS-FNA and the combined procedure was not specified [14].

Most studies did not evaluate diagnoses of malignancy and staging separately as we did, but limited their analyses to staging. Our study showed that the contribution of EUS-TA was paramount for the diagnosis, with a significantly higher sensitivity of the combined procedure compared with EBUS-TBNA (*p* < 0.0001), but not with EUS-TA (*p* = 0.13). To give an example in which EUS showed a true positive for malignancy diagnosis (but false negative in the staging analysis): In a patient with a PET+ left lung lesion with MLNs, EUS-TA in the station 3p was malignant and, therefore, staged N2. EBUS-TBNA, carried out at the same time in the station 10R (not reached with EUS), was also malignant, and thereby staged as N3 disease. This kind of situation occurred several times to the advantage of one or another technique depending on the location of the lung lesion with contralateral LNs. This is inherent to the technique used. This partly explains the less interesting results of the staging of EBUS-TBNA and EUS-TA when analyzed separately in our series.

We also showed that, after combined EBUS-EUS, the N status was upstaged in five patients compared with EBUS-TBNA or EUS-TA alone, from PET-CT staging. In these patients, the pejorative prognostic implications would have been major if both techniques had not been combined [15].

There are some technical differences in our study compared to data in the literature. In our series, EBUS-TBNA was performed in patients under general anesthesia with laryngeal masks in order to have optimal access for the puncture of upper paratracheal LNs (2R and 2L and, to a lesser extent, 4R and 4L), and to avoid the constraints related to ventilation tubes. The use of a laryngeal mask is not described in the literature in this setting. Compared to most series in which EBUS and EUS were performed with the same EBUS scope and operator [13,14,15], our EBUS was performed with a dedicated scope and an experienced pulmonologist, and the EUS procedure was performed with a dedicated EUS scope and an experienced digestive endosonographer. One of the most important differences with the previous series is the use of FNB needles in the majority of our patients (64.5%). This might explain the better performances of our EUS-TA results in the combined setting, although a randomized controlled study did not show statistically significant differences in terms of diagnostic yield or accuracy [16].

Why is staging with combined procedures still suboptimal, with a correct assessment in 74% with EBUS-TBNA, 68% with EUS-TA, and 85% with the combined procedure?

Both techniques underestimate the N stage in case of malignant supraclavicular LNs (three patients in our series).As mentioned above, stations 2R and 4R are not routinely punctured with EUS-TA owing to their right and anterior locations with the interposition of the trachea, except when LNs are above 20 mm in size. In our series, we reached LNs in the 4R station with EUS-TA in nine patients, with diagnosis and staging obtained in eight patients. In one patient with an LN in the 2R station which measured more than 20 mm and was punctured with EUS-TA, the diagnosis obtained was confirmed by mediastinoscopy, but the staging was established with EBUS-TBNA and could not be obtained with EUS-TA because it was an 11R station (right interlobar station) specific to EBUS-TBNA. LNs in the 2R and 4R stations were reached, whatever the size, with EBUS-TBNA in 6 and 64 patients, respectively (Table 2).Optimal EUS-TA including fanning was not feasible for the 2R (one patient) and 4R (nine patients) stations, since there is an interposition of the trachea and, therefore, the path of the needle passes along the right edge of the trachea to reach these anterior mediastinal stations. In two series, the MLN in the 2R and 4R stations were only ones accessed with EUS-TA in one patient [13,14]. In another series of 110 patients, EUS-TA was carried out in 2R and 4R stations in 10 and 12 patients, respectively [11]. However, no details on the size of the MLNs, the constraints of EUS-TA in these stations, or their consequences on the diagnosis and staging were mentioned [11,12,13,14].In our series, EUS staging was, however, clearly disadvantaged when compared with EBUS staging in this population, since they were mostly patients with right pulmonary lesions and N2 statuses corresponding to a majority of right and anterior MLNs that were not easily accessible by EUS-TA (in 40 patients with N2 disease, 27 had a right pulmonary lesion) (Table 1).Other stations are not easy or impossible to access by EUS or EBUS, such as stations 5 (sub-aortic, pulmonary-aortic window) and 6 (para-aortic), owing to the anatomical constraints of the aortic arch. In our study, EUS-TA was performed in station 5 in two patients without traversing the aorta, but not in station 6. These stations are better accessed with video-assisted thoracic surgery (VATS), which can reach almost every mediastinal LN station, especially stations 5 and 6, by means of left VATS. LNs in stations 5 and 6 cannot be reached by routine mediastinoscopy. Some authors have described transaortic puncture of LNs in station 6 in patients without serious complications [17,18]. Molina et al. described transvascular EBUS or EUS puncture through the aorta and the pulmonary artery to reach inaccessible mediastinal and hilar LNs and lung lesions, with an overall sensitivity of 71.5% and accuracy of 74.5% for diagnosing malignancy [19]. One can reasonably wonder about the risk of hematogenous tumor seeding [17,18,19]. Another report by Liberman et al. showed an EUS technique to puncture LNs in station 6 without traversing the aorta and without complications [20]. We did not use this technique because it is not a routine procedure.

Concerning the complication rate, we experienced a pleural empyema after lung puncture with EUS-FNB, which was drained under CT scan and treated with antibiotics in a patient whose course was favorable, without negative consequences on the management of lung cancer. We did not observe complications during EBUS-TBNA. In the literature, rare situations with mediastinitis and/or mediastinal abscess formation, often in cases of cystic or necrotic MLNs, have been described [21,22], including fatal outcomes in patients with poor clinical statuses [23,24]. In a systematic review and meta-analysis, no major complications after EUS-FNA for MLN and intrapulmonary lesions were described [25,26]. In a large series of 3123 patients, major complications such as mediastinal abscess and pneumomediastinitis were encountered in two patients after EBUS-TBNA [27]. In another study, EUS-FNA carried out in 213 patients with MLN caused esophageal perforation in 2 patients discharged after conservative management [28].

In NSCLC, European guidelines recommend at least three mediastinal stations to be assessed, including station 7, for SLA [29]. American guidelines recommend the assessment of at least three N2 stations for surgical staging [30]. Some data support that reasonable nodal status assessment could include at least 16 LNs. But, the minimal number of resected LNs seems to be controversial, since LNs could be fragmented. Therefore, en bloc resection should be attempted with the utmost care [31]. Indeed, a less invasive lymphadenectomy could downstage the real N staging. In our series, at least 16 LNs and 3 N2 stations were assessed by means of SLA.

A recent randomized controlled trial demonstrated that the strategy combining EBUS-EUS and surgical staging in potentially resectable lung cancer was superior to surgical staging alone in terms of clinical accuracy and cost-effectiveness [32].

The limitations of our study are mainly related to the retrospective design, but are also based on a prospective registry. There was bias linked to the recruitment of mainly patients with right pulmonary lesions and ipsilateral LNs, which may have disadvantaged the role of EUS-TA. The strengths are the use of dedicated scopes, experienced endoscopists; experienced nurses in both the fields of gastrointestinal and bronchopulmonary endoscopy; and appropriate gold standards such as VAM, which improves the visualization of LNs and SLA with VATS, allowing for extensive lymphadenectomy. Moreover, optimal preoperative diagnosis and staging in lung cancer require a strong collaboration between pulmonologists, gastroenterologists, thoracic surgeons, anesthesiologists, and histopathologists.

## 5. Conclusions

Our study demonstrates the significance of combining EBUS-TBNA and EUS-TA for the diagnosis and staging of MLNs in lung cancer patients, and confirms the significant role played by EUS-TA in the combined EBUS-EUS strategy for the diagnosis of malignancy. Gastroenterologists should, therefore, still be strong collaborators of pulmonologists, ideally for the sake of patients’ comfort, in a single combined procedure under sedation.

Mediastinal diagnosis and staging in lung cancer by combined EBUS-EUS in first intention could be easily organized and performed in outpatients. Furthermore, this strategy of combined endosonographic procedures is less invasive and more interesting in term of cost-effectiveness than mediastinoscopy, and avoids unnecessary mediastinoscopy and thoracotomy if EBUS or EUS is performed alone.

Finally, systematic combined EBUS-EUS represents an essential step in the diagnosis and staging of lung cancer, and furthermore has major therapeutic implications, such as targeted therapies, as it can obtain a better diagnostic yield for molecular analysis.

## Figures and Tables

**Figure 1 cancers-16-00728-f001:**
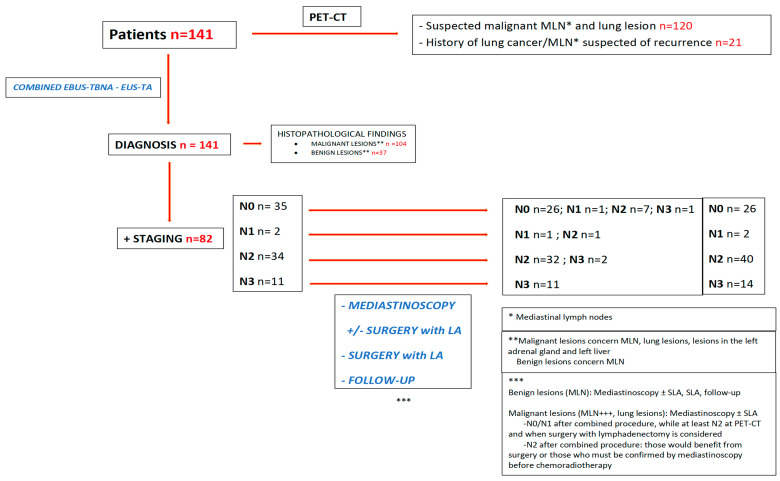
Flowchart.

**Figure 2 cancers-16-00728-f002:**
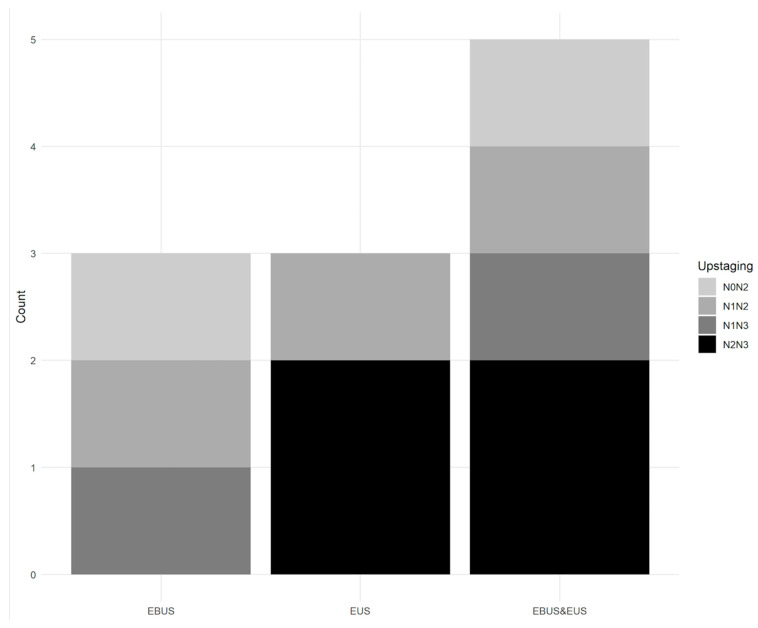
Added value of N upstaging to higher N stage of EBUS-TBNA alone, EUS-TA alone, and combined EBUS-TBNA-EUS-TA from PET-CT data.

**Table 1 cancers-16-00728-t001:** Patients and lesion characteristics.

Number of patients		141	
Age, median (range), years		66 (47–85)	
Sex ratio M/F		105/36	
Primary lesion location (including 6 patients with 2 pulmonary lesions)			
Right upper lobe		43	
Right middle lobe		7	
Right lower lobe		39	
Left upper lobe		34	
Left lower lobe		24	
Histological findings (combined EBUS-TBNA-EUS-TA)			
NSCLC		85	
Adenocarcinoma		48	
Squamous cell carcinoma (SCC)		29	
Indeterminate carcinoma		8	
NOS		4	
Little differentiated		4	
SCLC		17	
Composite lung cancer (SCC and SCLC)		1	
Neuroendocrine tumor		1	
Benign lesions		37	
Final staging (n = 82 patients)			
Combined EBUS-TBNA and EUS-TA		Mediastinoscopy/SLA/Follow-up	
N0	35	N0	26
N1	2	N1	2
N2	34	N2	40
N3	11	N3	14
Proportion of right/left pulmonary lesions in N2 population		27/13	

**Table 2 cancers-16-00728-t002:** Number of lymph node stations and other sites targeted by EBUS and EUS in 141 patients.

Stations	2R	2L	3	4R	4L	5	7	8	9	10R	10L	11R	11L	Lung Mass	LAG *	LL **
EBUS-TBNA	6	1	1 (3p)	64	19		87			14	5	26	17	4		
EUS-TA	1	5	3 (3p)	9	49	2	94	9	3					6	21	1

* Left adrenal gland, ** left liver.

**Table 3 cancers-16-00728-t003:** Diagnosis results in 141 patients between EBUS-TBNA, EUS-TA, and combined EBUS-TBNA and EUS-TA (combined procedure).

Tests	EBUS-TBNA	EUS-TA	Combined EBUS-TBNA and EUS-TA
Sensitivity *, % (95% CI)	75 [66–83]	87 [79–93]	93 [86–97]
Specificity, % (95% CI)	100 [90–100]	100 [91–100]	100 [89–100]
Accuracy, % (95% CI)	82 [74–88]	91 [85–95]	94 [89–98]
PPV, % (95% CI)	100 [95–100]	100 [96–100]	100 [96–100]
NPV, % (95% CI)	58 [45–70]	75 [61–86]	80 [64–91]

PPV: positive predictive value; NPV: negative predictive value; CI: confidence interval; EBUS-TBNA: endobronchial ultrasound-guided transbronchial needle aspiration; EUS-TA: endoscopic ultrasound-guided tissue acquisition; * EBUS-TBNA vs. combined approach, *p* = 0.000062; EUS-TA vs. combined approach, *p* = 0.13; McNemar’s test.

**Table 4 cancers-16-00728-t004:** Final staging results in 82 patients with EBUS-TBNA, EUS-TA, and combined EBUS-TBNA and EUS-TA (combined procedure).

Tests	EBUS-TBNA	EUS-TA	Combined EBUS-TBNA and EUS-TA
Sensitivity *, % (95% CI)	62 [49–75]	54 [40–67]	79 [66–88]
Specificity, % (95% CI)	100 [87–100]	100 [87–100]	100 [87–100]
Accuracy, % (95% CI)	74 [64–83]	68 [57–78]	85 [76–92]
PPV, % (95% CI)	100 [90–100]	100 [88–100]	100 [92–100]
NPV, % (95% CI)	55 [40–70]	50 [36–64]	68 [51–82]

PPV: positive predictive value; NPV: negative predictive value; CI: confidence interval; EBUS-TBNA: endobronchial ultrasound-guided transbronchial needle aspiration; EUS-TA: endoscopic ultrasound-guided tissue acquisition; * EBUS-TBNA vs. combined approach, *p* = 0.0077; EUS-TA vs. combined approach, *p* = 0.00051; McNemar’s test.

## Data Availability

The data presented in this study are available upon request from the corresponding author.

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
