# Peer review of "Improved Accuracy and Sensitivity in Diagnosis and Staging of Lung Cancer with Systematic and Combined Endobronchial and Endoscopic Ultrasound (EBUS-EUS): Experience from a Tertiary Center"

_cancers, 2024, doi:10.3390/cancers16040728_

Round 1

Reviewer 1 Report

Comments and Suggestions for Authors

Thank you for the opportunity to review this very interesting manuscript regarding mediastinal lymph node diagnosis.

 This study suggests the usefulness of a combined procedure of TBUS-TBNA and TUS-TA.

I would like to ask you a few questions regarding the content to deepen your understanding.

1.Your study design is very complex and difficult to understand.

In particular, the difference between the diagnostic phase and the final staging phase is not clear to me.

Are the 82 cases in the staging phase cases where the same lymph nodes were approached with EBUS-TBNA and EUS-TA?

I would like to see a clearer statement on these 82 cases.

2.How do you treat cases that are finally diagnosed as benign with regard to diagnostic yileds? Are all cases true negatives?

If yes, what tissue and cell samples were taken?

 Have you received a specific diagnosis, such as sarcoidosis or Castleman disease?

Is it correct to assume a true negative result simply because there are no malignant cells?

How long did you observe it and the final diagnosis was benign?

 3.In studies on diagnostic rates by bronchoscopy, the specificity and PPV are usually 100%. I therefore consider it sufficient to discuss sensitivity alone.

 4.This question relates to 12 cases where the combined EBUS-TBNA and EUS-TA procedure was performed in a tow procedure.

Was the examination process in these 12 cases followed by EUS-TA because of a negative EBUS-TBNA?

Please reconsider whether to include these 12 cases in the present analysis.

Author Response

We thank the reviewer 1 for its comments.

1.Your study design is very complex and difficult to understand.

In particular, the difference between the diagnostic phase and the final staging phase is not clear to me.

Are the 82 cases in the staging phase cases where the same lymph nodes were approached with EBUS-TBNA and EUS-TA?

I would like to see a clearer statement on these 82 cases.

We tried to make this study as understanding as possible. We separated as clearly as possible the diagnostic and the staging in the text (in Material and Methods paragraph with the flowchart (Figure 1)): the diagnostic phase concerns the 141 patients while 82 patients from them underwent an additional staging; the 59 remaining patients having been diagnosed as metastatic did not benefit from staging (paragraph results, lines 35-36).

Following your comments, we decided to modify the design of the flowchart regarding the diagnosis and staging in order to make it more understandable.

We tried to put forward the inseparable character of diagnosis in lung cancer (type of lung cancer) which is obtained mainly by endosonographic puncture of mediastinal lymph nodes (sometimes of lung mass, left liver lobe and left adrenal gland), and mediastinal staging in case of non-metastatic disease in lung cancer with tissue acquisition of mediastinal lymph nodes by combined EBUS-EUS after almost complete review of the stations in the anterior mediastinum (EBUS) and the posterior mediastinum (EUS).

Same and different mediastinal stations were reached by EBUS-TBNA and EUS-TA. That’s the reason why EBUS and EUS are complementary. For example, mediastinal lymph nodes in station 7 are reached by both techniques. Nevertheless, some stations are specific from one or the other technique (for example station 11 by EBUS and station 8 by EUS) and, in other cases, some stations are more easily achieved by one technique rather than another. The other advantage to puncture the same stations with both techniques is to improve the diagnostic yield and then to obtain optimal histopathological samples not only for diagnosis, but also for molecular analysis.

2.How do you treat cases that are finally diagnosed as benign with regard to diagnostic yields? Are all cases true negatives?

If yes, what tissue and cell samples were taken?

Have you received a specific diagnosis, such as sarcoidosis or Castleman disease?

Is it correct to assume a true negative result simply because there are no malignant cells?

How long did you observe it and the final diagnosis was benign?

All cases are related to lung cancer, there are no other diseases such as sarcoidosis, etc…(Table 1).

The benign or malignant character concerns results of punctures for diagnosis of lung cancer in patients finally diagnosed with lung cancer. The diagnosis of malignancy or benignity of these mediastinal lymph nodes punctured by combined EUS-TBNA/EUS-TA determines the staging N0, N1, N2 or N3 diseases. Benign results for malignancy after combined procedure for staging were confirmed by mediastinoscopy, SLA or rarely by follow-up and results of final staging N0, N1, N2 and N3 are reported in the new flowchart (Fig. 1). When benign results are confirmed, they are considered as true negative (cf page 6, paragraph Statistical analysis, lines 17-28).

3.In studies on diagnostic rates by bronchoscopy, the specificity and PPV are usually 100%. I therefore consider it sufficient to discuss sensitivity alone.

We reported all the statistic values in tables to be complete and systematic. We agree with you that the main indicator of performance in this setting is sensitivity (but also accuracy and NPV), and we precise it for diagnosis and staging in the text (page 7, results, lines 51-59).

4.This question relates to 12 cases where the combined EBUS- TBNA and EUS-TA procedure was performed in a tow procedure.

Was the examination process in these 12 cases followed by EUS-TA because of a negative EBUS-TBNA?

Please reconsider whether to include these 12 cases in the present analysis.

These 12 cases are related to the beginning of our combined procedures. In these cases, EBUS and EUS were systematically performed but not in the same time (mean period between procedures 14 days). All other combined endosonographic procedures were performed in a single step anesthesia.

We did not realize the EUS-TA because of negative EBUS-TBNA but because this was performed within a framework of a systematic setting. We need the anterior mediastinal staging of the EBUS-TBNA and the posterior mediastinal staging of the EUS-TA to determine precisely the N status of the patient (staging), to avoid unnecessary mediastinoscopy and thoracotomy.

Reviewer 2 Report

Comments and Suggestions for Authors

Diagnosis and staging are pivotal in patient management. A challenge has always been the adequate sampling from mediastinal lymph nodes. This study compares Endobronchial and endoscopic ultrasounds. Included patients have had both procedures done so they serve as their own control and the design is therefore very valid compared to comparison of cohorts that have had either one or the procedure done. The material is well described. Standard statistical analysis was used to estimate sensitivity and specificity and related outcomes.

The results section gives valuable information on localisation of lesions biopsied.

Superior outcomes were noted for the combined procedure.

The discussion is well balanced and discusses the challenges that individual cases represent.

The authors suggest that gastroenterologist should be involved to perform EUS-TA but the question is whether a pulmonologist should train in this procedure to make the diagnostic work flow more robust.

Author Response

We thank the reviewer 2 for its comments.

Diagnosis and staging are pivotal in patient management. A challenge has always been the adequate sampling from mediastinal lymph nodes. This study compares Endobronchial and endoscopic ultrasounds. Included patients have had both procedures done so they serve as their own control and the design is therefore very valid compared to comparison of cohorts that have had either one or the procedure done. The material is well described. Standard statistical analysis was used to estimate sensitivity and specificity and related outcomes.

The results section gives valuable information on localisation of lesions biopsied.

Superior outcomes were noted for the combined procedure.

The discussion is well balanced and discusses the challenges that individual cases represent.

The authors suggest that gastroenterologist should be involved to perform EUS-TA but the question is whether a pulmonologist should train in this procedure to make the diagnostic work flow more robust.

We suggest that the gastroenterologists should be involved in lung cancer diagnosis and mainly in lung cancer staging:  gastroenterologists who perform endoscopic ultrasound are often confronted with the assessment of mediastinal lymph nodes from various origin as lung cancer, esophageal carcinoma, lymphoma, sarcoidosis… but they are not used to the complementarity of EUS and EBUS in lung cancer staging and to the diagnostic yield improvement with both techniques (not only for the diagnosis but also for molecular analysis). It is important to sensitize them on this topic.

As a reminder, the European Society of Gastrointestinal Endoscopy (ESGE) launched in 2015 guidelines on combined EBUS-EUS in lung cancer diagnosis and staging, in association with the European Respiratory Society (ERS) and the European Society of Thoracic Surgeons (ESTS)1.

In our center and since many years, the combined EBUS-EUS procedures are performed by a gastroenterologist for EUS and a pulmonologist for EBUS in patients with lung cancer in a single step procedure with only one sedation or general anesthesia. This is facilitated by the fact that the nurses are involved both in bronchopulmonary and digestive endoscopy. Nevertheless, in some centers, some pulmonologists are used to perform both EBUS and EUS with the same scope in lung cancer diagnosis and staging (EBUS and EUS-B). Obviously, pulmonologists could train in EUS with EBUS scope (EUS-B) or EUS scope.

1Vilmann, P.; Clementsen, P.F.; Colella, S.; Siemsen, M.; De Leyn, P.; Dumonceau, J.M.; Herth, F.J.; Larghi, A.; Vazquez-Sequeiros, E.; Hassan, C.; et al. Combined endobronchial and esophageal endosonography for the diagnosis and staging of lung cancer: European Society of Gastrointestinal Endoscopy (ESGE) Guideline, in cooperation with the European Respiratory Society (ERS) and the European Society of Thoracic Surgeons (ESTS). Endoscopy 2015, 47, c1,

Reviewer 3 Report

Comments and Suggestions for Authors

Important and necessary research has been carried out. The research process and results are described in great detail.

However, the text is quite difficult to read (follow) due to the chosen presentation of the material. In my opinion, it would be easier to read and would add to the value of the study, if the material was presented in a slightly different way.

For example, the study could be limited to lung cancer patients only. In this case, the clinical situation would be clearly defined, as is usually the case in clinical practice. Confirmation of lung cancer (first step), staging (second step), and accuracy of staging verified by a reference method (third step).

If necessary, non-malignant cases are considered separately.

Now when lung cancer and other cases are presented in parallel, it is difficult to assess the results obtained (sensitivity, specificity, positive and negative predictive values), which partly contradict each other.

Author Response

We thank the reviewer 3 for its comments.

Important and necessary research has been carried out. The research process and results are described in great detail. However, the text is quite difficult to read (follow) due to the chosen presentation of the material. In my opinion, it would be easier to read and would add to the value of the study, if the material was presented in a slightly different way.

For example, the study could be limited to lung cancer patients only. In this case, the clinical situation would be clearly defined, as is usually the case in clinical practice. Confirmation of lung cancer (first step), staging (second step), and accuracy of staging verified by a reference method (third step).

If necessary, non-malignant cases are considered separately. Now when lung cancer and other cases are presented in parallel, it is difficult to assess the results obtained (sensitivity, specificity, positive and negative predictive values), which partly contradict each other.

Our study is only limited to lung cancer patients. The benign and malignant lesions reported in the Table 1 (Patient and characteristics) concern the lesions (mainly lymph nodes) punctured by combined EBUS-EUS for diagnosis in lung cancer patients.

The diagnostic phase concerns the 141 patients while 82 patients from them underwent an additional staging.

We understand that the lecture of the flowchart could be confused. Following your comments, we decided to modify the design of the flowchart regarding the diagnosis and staging.

Round 2

Reviewer 1 Report

Comments and Suggestions for Authors

Again, this question concerns 12 cases in which combined EBUS-TBNA and EUS-TA were performed in a two procedure.

Did you perform EUS in these 12 cases after confirming that the tissue diagnosis was negative by EBUS? Or did you plan to perform EUS from the beginning regardless of the EBUS result?

The implication of this question is that performing EUS on cases with negative EBUS results will only result in EUS providing additional value.

Author Response

Thank you for your interesting questions and comments.

Regardless of whether lymph nodes were negative or positive for malignancy, we performed the procedures related to these 12 cases in a systematic combined setting. The combined EBUS-EUS related to these 12 cases were only performed at the beginning of our study. At this period, the exam slots of a single step procedures were not sufficient to manage all the cases of mediastinal staging in lung cancer. Indeed, our combined EBUS-EUS procedures have been and continue to be performed in the endoscopy program of the gastroenterologist who performed the EUS procedure.

We fully agree that performing EUS on cases with negative lymph nodes for malignancy after the EBUS procedure certainly results in an additional value of EUS, and conversely.

Following your comments, we decided to add a sentence related to this aspect in the "Results" paragraph.

Reviewer 3 Report

Comments and Suggestions for Authors

None

Author Response

Thank you for your strong interest in our article